# A High-Linearity Closed-Loop Capacitive Micro-Accelerometer Based on Ring-Diode Capacitance Detection

**DOI:** 10.3390/s23031568

**Published:** 2023-02-01

**Authors:** Qi Tao, Bin Tang

**Affiliations:** China Academy of Engineering Physics, Institute of Electronic Engineering, Mianyang 621000, China

**Keywords:** capacitive accelerometer, closed-loop, low non-linearity, MEMS

## Abstract

This paper presents an implementation scheme and experimental evaluation of a high-linearity closed-loop capacitive accelerometer based on ring-diode capacitance detection. By deducing the capacitance detection model of the ring-diode considering the influence of the diode, the existing theoretical model error of the ring-diode is corrected and a closed-loop scheme of reusing the detection electrode and the control electrode of the MEMS die is designed to apply this detection scheme to the parallel-plate accelerometer, which only has three independent electrodes. After analyzing the non-linear problems in the existing closed-loop control schemes, a theoretically absolute linear closed-loop control scheme is proposed, and an integrated closed-loop accelerometer is realized by combining the closed-loop diode detection. The experimental results of the ring-diode detection model are in agreement with the theoretical formula. The non-linearity of the accelerometer within ±1 g after the closed-loop is 130 ppm, compared with 1500 ppm when the open-loop is used.

## 1. Introduction

The capacitive accelerometer can detect the capacitance change caused by proof mass displacement under the action of inertia force and solve the acceleration information. The MEMS capacitive accelerometer, with the characteristics of low cost, high precision, low power consumption, and small size [1], is widely used in various environments [2,3]. Due to its small size, the capacitance of the MEMS capacitive accelerometer is in the order of pF, and the capacitance detection accuracy needs to reach the order of aF [4], which strives for greater requirements for the signal-to-noise ratio of the sensitive structure and the detection circuit. The capacitive open-loop accelerometer has non-linearity in acceleration measurement due to the non-linearity in capacitance detection, and the low stiffness and large range of the open-loop cannot be met at the same time, which makes it difficult to continue to improve the performance of the accelerometer. As a result, it is necessary to introduce closed-loop control to improve the linearity and range of the accelerometer [5].

In terms of capacitance detection, the ring-diode capacitance detection circuit, as a simple passive capacitance–voltage conversion scheme, has high gain, and the output is linearly related to the detection capacitance difference [6,7]. It can be read directly by ADC without introducing a two-stage amplification circuit. However, there is much error of the original theoretical formula when the detection capacitance is as small as pF~fF. The existing closed-loop control methods based on the ring-diode need to introduce new independent electrodes, which cannot be better applied to all capacitive accelerometers.

The performance of the closed-loop capacitive accelerometer partly depends on the closed-loop control scheme [8]. The existing closed-loop capacitive accelerometer control schemes have a non-linear term that cannot be eliminated. In actual products, the corrected non-linearity is basically more than 300 ppm [9,10], and only AXO315 [11] with an in-plane structure scheme achieves 100 ppm. The non-linearity that cannot be further reduced affects the improvement of the capacitive accelerometer’s performance [12].

In this paper, a closed-loop capacitive accelerometer measurement and control scheme is proposed, which uses a ring-diode detection scheme for capacitance detection and adopts a high-linearity closed-loop control scheme. This scheme has a simple structure and can obtain high linearity, which can be applied to differential capacitive accelerometers of any variable-gap capacitor.

## 2. Measurement and Control Scheme of Capacitive Accelerometer

### 2.1. Analysis of Working Principle of Capacitive Accelerometer

Two common structures of the MEMS capacitive accelerometer are shown in Figure 1. The capacitive accelerometer converts the displacement of the proof mass under the action of inertia force into the capacitance change and obtains the acceleration by detecting the capacitance change [13,14]. Figure 1a shows the structure of the variable-gap capacitor accelerometer, and Figure 1b shows the structure of the variable-area capacitive accelerometer. One of the variable-gap capacitor accelerometers is a comb structure composed of several smaller plates, and the other is a parallel-plate structure composed of three larger parallel plates, while the variable-area capacitive accelerometer is usually a comb structure composed of several smaller plates. As a capacitive accelerometer, the parallel-plate structure composed of larger plates can usually achieve much higher sensitivity than the variable-area capacitive accelerometer [15].

Figure 2a shows the structure of the parallel-plate accelerometer, which is composed of a top electrode, bottom electrode, and middle electrode, and the middle electrode can move under the action of inertial force. The distance between the middle electrode and the top electrode is d1, and the capacitance is C1, while the distance between the middle electrode and the bottom electrode is d2, and the capacitance formed is C2. Assuming that the distance and capacitance between the middle electrode and another electrode are d and C, the acceleration is a, the inertial force acting on the proof mass is F, the displacement of the proof mass under the inertial force is x, and the change of capacitance, C, is ∆*C*. As shown in Figure 2b, inertia force is converted into displacement, displacement is converted into capacitance change, and the conversion relationship is:(1)$$\frac{∆C}{a}=\frac{2\varepsilon A}{d^{2}-x^{2}}*\frac{m}{k}$$
where A and ε are the area of the capacitor plate and the dielectric constant, and m and k are the proof mass and the stiffness of the sensitive structure, respectively.

It can be seen from Formula (1) that under the same capacitance detection accuracy, a larger sensitive proof mass, m, a smaller stiffness, k, and a smaller spacing, d, are required to obtain a higher acceleration detection accuracy. When k is small, the displacement, x, is limited by the spacing, d, resulting in a smaller range of acceleration during open-loop detection ($a<\frac{kd}{m}$). Due to the non-linearity introduced by capacitance during $\frac{∆C}{a}$ conversion, when x gradually increases to close to d, the non-linearity becomes larger and larger. Therefore, the open-loop detection cannot take the detection accuracy, linearity, and range into consideration at the same time. Thus, it is necessary to introduce closed-loop control to stabilize the mass block at the set position to achieve high precision, large range, and high linearity.

As shown in Figure 2c, to counteract the inertial force, the electrostatic feedback force, $F_{f}$, for the closed-loop detection system is introduced; as a result, the proof mass is kept at a fixed position. The feedback force $F_{f}=ma=\frac{2\varepsilon AV_{f}{}^{2}}{d^{2}}$, and the detected acceleration $a=\frac{F_{f}}{m}=\frac{2\varepsilon AV_{f}{}^{2}}{md^{2}}$, and the acceleration detection has no connection with the stiffness, k, in the accelerometer structure. Therefore, during the structural design operation, the stiffness, k, can be designed to be very low without affecting the accelerometer range. According to Formula (1), the lower the stiffness, the higher the accelerometer detection sensitivity. The feedback force is introduced to keep the proof mass at a fixed position, which solves the non-linear problem caused by the change in capacitor electrode spacing and the capacitance change in the open-loop control. However, in closed-loop control, new non-linear problems will be caused due to inaccurate electrode spacing and proof mass position, which will affect the further improvement of the performance of the MEMS capacitive accelerometer.

The closed-loop capacitive structure usually adopts a differential detection structure to ensure linearity. Therefore, a differential detection electrode and control electrode are required in the closed-loop accelerometer, and at least three electrodes are required to meet the detection and control requirements. Common practices include the detection or control electrode reuse scheme of Colibrys Aida [9] and the separation scheme of the detection electrode and control electrode of Physical Logic Ltd. [10]. To obtain a higher sensitivity and a larger range of the accelerometer, its proof mass, detection electrode capacitance area, and control electrode capacitance area need to be increased as much as possible [16]. The separation of the detection electrode and the control electrode leads to the reduction of the detection capacitance area and the control capacitance area compared with the reuse scheme, which reduces the sensitivity and the range of the accelerometer. At present, high-precision accelerometers include out-of-plane accelerometers with a parallel-plate structure composed of three layers of bonding (Colibrys [17], Northrop Grumman LITEF [18]), as shown in Figure 2a, and an in-plane accelerometer scheme composed of a comb structure (Tronics [11], Physical Logic Ltd. [19]), as shown in Figure 1. The out-of-plane bonding as shown in Figure 2a can accurately control the capacitance gap at about 2 um. Due to the deep reactive ion etching (DRIE) process, it is difficult to decrease the capacitance gap of the in-plane comb structure, as shown in Figure 1, to be less than 3 um when the etching depth is large [15,20]. Therefore, it is more difficult to achieve higher sensitivity and range at the same time in the in-plane comb structure than that in the out-of-plane structure. The advantages of the in-plane comb structure lie in the higher linearity coming from the translation of the electrode as well as the ease of fabricating separate control electrodes and detection electrodes. After adopting the parallel-plate structure, which can achieve high sensitivity and a large range, and deeply exploring the closed-loop control scheme of the three-electrode structure and the non-linear problem it produced, a closed-loop control scheme with high linearity is presented in this paper to solve the non-linear problem of the parallel-plate structure.

### 2.2. Closed-Loop Detection and Control Scheme

The original ring-diode capacitance detection scheme needs two differential detection electrodes and one carrier input electrode when applied to the accelerometer. The commonly used MEMS capacitive accelerometer (such as the accelerometer of Colibrys [16]) adopts a parallel-plate structure to obtain a great capacitance area and sensitivity quality [15]. It has only one common electrode and two differential electrodes. If closed-loop control is to be realized, there is no independent electrode input control quantity. Therefore, the original ring-diode detection circuit as shown in Figure 3 is improved in this paper, so that the differential electrode of the accelerometer can realize detection and control multiplexing. The schematic diagram is shown in Figure 4.

As shown in Figure 4, the square wave source, $U_{S}$, generates a 4 MHz square wave signal, which can enter the ring-diode through the high-pass filter after passing through the MEMS die. While the voltage frequency used for closed-loop control is below 4 kHz, the low-frequency control signal can be applied to the same electrode with which the high-frequency detection signal is used at the same time, and then the high-pass filter is used during the subsequent detection to remove the applied low-frequency control signal. Consequently, the control and detection of a shared electrode can be achieved. Capacitive detection is realized by taking advantage of the high amplification and simple structure of the ring-diode, then the closed-loop circuit is realized based on the ring-diode without adding an independent electrode to the accelerometer, to achieve the closed-loop capacitive accelerometer scheme with a high signal-to-noise ratio, simple structure, and large range. Table 1 shows the parameters in the closed-loop control circuit.

The control loop uses MCU to read ADC data and process them according to the high-linearity control scheme derived later, and then acts on the upper and lower electrodes of the accelerometer through dual DAC to achieve closed-loop control of the accelerometer.

### 2.3. Closed-Loop Capacitive Accelerometer

The complete accelerometer structure is shown in Figure 5, including DCDC and LDO power supply modules, a detection carrier module, MEMS die, a ring-diode CV conversion module, a MCU data processing module with 24-bit ∑-Δ ADC and a temperature sensor, and a 16-bit dual-channel DAC module. The input single power supply ranges from 2.5 to 16 V, and the accelerator output adopts 3.3 V SPI. The physical structure size is 22 × 23 mm, and the power consumption is 150 mw. The physical picture is shown in Figure 6.

The MEMS die is based on the out-of-plane 3-stack silicon wafer structure, as shown in Figure 7, which is constituted of a proof mass with a surface of 3.2 mm^2^ and a thickness of 380 microns, attached by a silicon spring to a frame and separated from the detection plates on each side by a thin oxide of 2 microns. The capacitance formed between the middle electrode and the top or the bottom electrode is about 30 pf, and the resonant frequency of the MEMS die is about 831 Hz. The MEMS die is packaged with a universal ceramic shell, and meanwhile, DIP-14 packaging is used for the convenience of testing, but it occupies a large area. In subsequent actual production, the MEMS die can be packaged into a customized small shell to further reduce the structure size.

## 3. Systems Analysis

### 3.1. Derivation of Accurate Model of Ring-Diode Capacitance Detection Circuit

Based on the original research, the theoretical formula derivation of the ring-diode in this paper takes the diode junction capacitance into consideration, and a more detailed theoretical formula derivation of the ring-diode can be obtained from existing research [6,21,22].

As shown in Figure 3, the amplitude of the square wave, $U_{s}$, is $U_{m}$. The junction capacitance of four identical diodes is $C_{D}$ and the voltage drop is $U_{D}$. The carrier period is T, which is longer than the time from charge or discharge to stabilization of $C_{1}$ and $C_{3}$.

Supposing there is no charge left on the capacitances at the initial stage, the DC voltage $U_{a}$ at point a and $U_{b}$ at point b will be stable during plenty of cycles of the square wave. Then, at the end of the positive period of the square wave and when all the capacitances have been charged or discharged to a stable state, the charge of the change across $C_{3}$ and $C_{4}$ can be solved as:(2)$$2\left(U_{b}-U_{a}+∆U_{b}-∆U_{a}\right)C_{D}+(2U_{m}-2U_{D}-U_{a}-∆U_{a}+U_{b}-∆U_{b})C_{1}\;=\frac{U_{a}+∆U_{a}}{2R_{b1}}T+2∆U_{a}C_{3}$$
(3)$$2\left(U_{a}-U_{b}+∆U_{a}-∆U_{b}\right)C_{D}+(2U_{m}-2U_{D}-U_{b}-∆U_{b}+U_{a}-∆U_{a})C_{2}\;=\frac{U_{b}+∆U_{b}}{2R_{b2}}T+2∆U_{b}C_{4}$$
where the change voltage of points a and *b* is $∆U_{a}$ and $∆U_{b}$.

At the end of the negative period of the square wave and when all the capacitances have been charged or discharged to a stable state, the charge of the change across $C_{3}$ and $C_{4}$ can be solved as:(4)$$2\left(U_{b}-U_{a}+∆U_{a}-∆U_{b}\right)C_{D}-(2U_{m}-2U_{D}+U_{a}-∆U_{a}-U_{b}-∆U_{b})C_{2}\;=\frac{U_{a}-∆U_{a}}{2R_{b1}}T-2∆U_{a}C_{3}$$
(5)$$2\left(U_{a}-U_{b}+∆U_{b}-∆U_{a}\right)C_{D}-(2U_{m}-2U_{D}+U_{b}-∆U_{b}-U_{a}-∆U_{a})C_{1}\;=\frac{U_{b}-∆U_{b}}{2R_{b2}}T-2∆U_{b}C_{4}$$
where the change voltage of points a and *b* is $-∆U_{a}$ and $-∆U_{b}$.

Generally, the capacitances $C_{3}$ and $C_{4}$ take the same value, *C*_*L*_, and the resistances $R_{b1}$ and $R_{b2}$ take the same value, $R_{b}$, and thus the output voltage, $U_{o}$, can be solved by combining the Formulas (2)–(5), as:(6)$$\{\begin{array}{c} U_{b}=-U_{a}\\ ∆U_{b}=∆U_{a} \end{array}$$
(7)$$U_{o}=2U_{a}=\frac{2(U_{m}-U_{D})\left(2C_{L}+\frac{T}{2R_{b}}\right)\left(C_{1}-C_{2}\right)}{(C_{1}+C_{2})\left(2C_{L}+4C_{D}+\frac{T}{R_{b}}\right)+\left(4C_{D}+\frac{T}{2R_{b}}\right)\left(2C_{L}+\frac{T}{2R_{b}}\right)+2C_{1}C_{2}}$$

It can be obtained from Formula (7) that when $C_{L}$ and $C_{1}$ are in the same order of magnitude, the diode capacitance has a great influence on the conversion relationship of detection capacitance and voltage. In practice, the diode capacitance is in the order of pF, and the diode junction capacitance used in this paper is 0.7 pF.

The open-loop test circuit diagram is shown in Figure 4. The ring-diode detection circuit is used to convert the capacitance change of the accelerometer into a voltage signal, which is directly read out through the differential ADC. Assuming that the capacitance changes linearly, the result is derived from the above formula, and the values of capacitance, $C_{1}$ and $C_{2}$, are assigned to be 36 and 32.5 pF, respectively, and the carrier amplitude $U_{m}=1V$. According to the above formula, the output voltage $U_{o}=78.5\;\mathrm{mV}$, while the circuit simulation result is 75.2 mV, and the actual test result is 82.5 mV. The result of the theoretical formula derivation is basically consistent with that from the simulation and the actual test. When the fixed capacitance is 34 pF, the capacitance difference voltage amplification factor is 23.6 mV/pF, which presents high gain, and thus there is no need to apply two-stage active amplification in the detection circuit. The introduction of new noise can be avoided, and in addition, the output voltage of ring-diode conversion can be directly connected to the differential ADC for voltage reading.

### 3.2. Derivation of Closed-Loop Control Scheme

The common high-precision closed-loop accelerometer sensing structure [23] is the differential capacitance structure shown in Figure 2a, which consists of a top electrode, middle electrode, and a bottom electrode. The middle electrode is usually used as the proof mass to sense the acceleration, and the top and bottom electrodes are fixed. The closed-loop capacitive accelerometer balances the inertial force through the electrostatic force generated by the voltage difference between the top electrode and the middle electrode and the voltage difference between the bottom electrode and the middle electrode. As a result, the proof mass is controlled at a fixed position to detect the acceleration.

Assuming that the voltage of the top electrode of the parallel-plate accelerometer is $U_{T}$, the voltage of the middle electrode is $U_{M}$, and the voltage of the bottom electrode is $U_{B}$. When the electrostatic force of closed-loop control and the inertial force are equal, the middle electrode is controlled at a fixed position, the distance between the middle electrode and the top electrode is $d_{1}$, and the distance between the middle electrode and the bottom electrode is $d_{2}$. Assuming that the acceleration is a, the plate area is A, the quality of proof mass is m, and the dielectric constant is ε. The equation of inertia force and electrostatic force can be obtained as:(8)$$\left(\frac{{\left(U_{T}-U_{M}\right)}^{2}}{d_{1}{}^{2}}-\frac{{\left(U_{B}-U_{M}\right)}^{2}}{d_{2}{}^{2}}\right)*\varepsilon A=ma$$
(9)$$\to a=\varepsilon A*\frac{U_{T}{}^{2}d_{2}{}^{2}-U_{B}{}^{2}d_{1}{}^{2}+U_{M}{}^{2}\left(d_{2}{}^{2}-d_{1}{}^{2}\right)-2U_{M}\left(d_{2}{}^{2}U_{T}-d_{1}{}^{2}U_{B}\right)}{md_{1}{}^{2}d_{2}{}^{2}}$$

The three electrodes can be applied with control voltage. The common control methods are as follows: (1) The voltage of the top electrode and the bottom electrode is fixed, and the voltage of the middle electrode is changed to offset the inertial force. (2) In the top and bottom electrodes, only a variable voltage is applied to one of them at the same time according to the acceleration, the voltage at the other side is fixed, and the voltage at the middle electrode is always fixed. (3) The voltages of the top and the bottom electrodes are changed simultaneously, and the voltage of the middle electrode is fixed. The effects of the three control modes are analyzed below.

#### 3.2.1. Control the Middle Electrode Voltage Only

The voltage of the top and bottom electrodes is fixed. To make the control symmetrical, the voltage of the top and bottom electrodes is simplified to take the opposite value, $U_{TB}=\left|U_{T}\right|=\left|U_{B}\right|$, and the middle electrode’s voltage is variable. Then, the relationship between acceleration and control quantity is:(10)$$a=\varepsilon A\frac{\left(U_{TB}{}^{2}+U_{M}{}^{2}\right)\left(d_{2}{}^{2}-d_{1}{}^{2}\right)-2U_{M}U_{TB}\left(d_{2}{}^{2}+d_{1}{}^{2}\right)}{md_{1}{}^{2}d_{2}{}^{2}}$$
(11)$$a=Constant+\varepsilon A\frac{U_{M}{}^{2}\left(d_{2}{}^{2}-d_{1}{}^{2}\right)-2U_{M}U_{TB}\left(d_{2}{}^{2}+d_{1}{}^{2}\right)}{md_{1}{}^{2}d_{2}{}^{2}}$$

In the existing closed-loop control scheme of the accelerometer, it is generally considered that $d_{2}=d_{1}$, and Formula (11) can be simplified as:(12)$$a=-2\varepsilon A\frac{U_{M}U_{TB}\left(d_{2}{}^{2}+d_{1}{}^{2}\right)}{md_{1}{}^{2}d_{2}{}^{2}}$$
(13)$$U_{M}=-a\frac{md_{1}{}^{2}d_{2}{}^{2}}{2\varepsilon A*U_{TB}\left(d_{2}{}^{2}+d_{1}{}^{2}\right)}$$

The deduced results, as shown in Formula (13), show that the relationship between acceleration, a, and control voltage, $U_{M}$, is linear.

However, in the actual control, the differential electrode of the accelerometer cannot be completely symmetrical during the processing, and because the detection circuit cannot be absolutely symmetrical during the differential detection, coupled with the influence of environmental factors such as temperature and stress, the closed-loop control of $d_{2}=d_{1}$ cannot be accurately achieved and the difference between $d_{2}$ and $d_{1}$ cannot be determined. Thus, the relationship between the obtained acceleration and the control quantity cannot be solved linearly, and the closed-loop control method cannot eliminate the non-linear term in principle. Although this control method is relatively simple, requiring only one control quantity, and the ideal state acceleration is directly linearly related to the control voltage, the non-linear items that cannot be eliminated in practice affect the closed-loop control effect.

#### 3.2.2. Control the Voltage of One Side Electrode Only

At the same time, there is one control voltage being applied to the top electrode or the bottom electrode (single-side control scheme of Colibrys Adia [9,24]) and the voltage of the middle electrode is fixed. Assume $U_{M}=0\;V$, and when the top electrode acts, the voltage of the bottom electrode $U_{B}=0\;V$. The relationship between acceleration and control quantity can be simplified as:(14)$$a=\varepsilon A\frac{U_{T}{}^{2}d_{2}{}^{2}}{md_{1}{}^{2}d_{2}{}^{2}}$$

The relationship between acceleration and control quantity, $U_{T}{}^{2}$, can be presented as $Scale_{T}=\varepsilon A\frac{1}{md_{1}{}^{2}}$. Similarly, when only the bottom electrodes act, the relationship between acceleration and the control quantity, $U_{B}{}^{2}$, can be presented as $Scale_{B}=-\varepsilon A\frac{1}{md_{2}{}^{2}}$. The voltage action of the top or bottom electrodes depends on the direction and magnitude of the acceleration. However, $d_{2}\neq d_{1}$, and the conversion of the acceleration direction results in the change of the scaling factor between acceleration and control quantity, which leads to two intersecting rays instead of a straight line in the graph of acceleration and control quantity, which can be shown as $Scale_{T}\neq Scale_{B}$. When two DACs are used to control the voltage of the top and bottom electrodes, respectively, the oscillation will occur during the switch of the control voltage.

#### 3.2.3. Control the Voltage of Top and Bottom Electrodes Simultaneously (Control Scheme Used in This Paper)

Assume that the middle electrode’s voltage $U_{M}=0\;V$, the top and bottom electrodes have variable voltage, and the relationship between acceleration and control quantity is:(15)$$a=\varepsilon A\frac{U_{T}{}^{2}d_{2}{}^{2}-U_{B}{}^{2}d_{1}{}^{2}}{md_{1}{}^{2}d_{2}{}^{2}}$$

Introduce common mode constant B and differential mode control quantity ∆B, and suppose $U_{T}{}^{2}=B+∆B$,$U_{B}{}^{2}=B-∆B$, and then the equation can be obtained as follows:(16)$$a=\varepsilon A\frac{B\left(d_{2}{}^{2}-d_{1}{}^{2}\right)+\left(d_{2}{}^{2}+d_{1}{}^{2}\right)∆B}{md_{1}{}^{2}d_{2}{}^{2}}=Constant+Scale*∆B$$

The scaling factor between the acceleration, a, and the control quantity, ∆B, is$\;Scale=\varepsilon A\frac{\left(d_{2}{}^{2}+d_{1}{}^{2}\right)}{md_{1}{}^{2}d_{2}{}^{2}}$, and the $Constant=\varepsilon AB\frac{\left(d_{2}{}^{2}-d_{1}{}^{2}\right)}{md_{1}{}^{2}d_{2}{}^{2}}$.

When B is taken as a constant, the control quantity, ∆B, is linearly related to the acceleration, a, the difference between $d_{2}$ and $d_{1}$ does not affect the linearity again.

Control quantity$\;∆B=U_{T}{}^{2}-B=B-U_{B}{}^{2}$, and the maximum control voltage available in actual control is $U_{MAX}=MAX\left(\left|U_{T}\right|\right)=MAX\left(\left|U_{B}\right|\right)$. To obtain a larger range in both positive and negative directions at the same time, the recommended value of B is $B=U_{MAX}{}^{2}/2$.

Two DACs are used in closed-loop control, including a power reference. Assuming that the full-range reference voltage is $v_{Ref}$, and the voltage of one DAC’s output is $U_{B}$, $U_{B}=\sqrt{B-∆B}$, the maximum voltage of DAC is $v_{Ref}$, and thus $B=v_{Ref}{}^{2}/2$ because the recommended value $B=U_{MAX}{}^{2}/2$; then, the digital control quantity, $DAC\_b_{digit}$, of the DAC is:(17)$$DAC\_b_{digit}=0xFFFFFFFF*\frac{U_{B}}{v_{Ref}}=0xFFFFFFFF*\sqrt{\left(\frac{1}{2}-\frac{∆B}{v_{Ref}{}^{2}}\right)}$$

According to Formula (17), the inaccuracy of the reference voltage, $v_{Ref}$, will only affect the proportion of ∆B and will not cause new non-linearity. In this paper, the DAC adopts a dual DAC scheme with single-chip integration and a common voltage reference to ensure the symmetry between the two DACs.

As shown in Figure 8a, in this control scheme, the linear quantity related to the acceleration is ∆B, but the relationship between the final output voltage and ∆B is shown in Formula (17), which requires a time-consuming square root operation. To save computing resources, improve the computing speed, and control the frequency in this closed-loop control, all the calculations are operated as integer numbers. Although the calculation speed is improved by using integral number operation, there are quantization errors in the calculation results, especially after the square root operation, such as $\left(int\right)\sqrt{65,535^{2}\sim \left(65,536^{2}-1\right)}=65,535$. This will cause errors between the control quantity, ∆B, and the closed-loop control result, $DAC\_b_{digit}$, and affect the linearity of the accelerometer. The theoretical acceleration output error of the quantization error before correction is shown in Figure 8b, and the maximum error is about 80 µg. The larger quantization error needs to be corrected to improve the performance of the accelerometer.

The quantization error caused by the closed-loop control operation is deduced as follows. In the process of converting the control quantity, ∆*B*, of Formula (17) to the control voltage, U, supposing that the square, $U_{T}{}^{2}$, of the top electrode output voltage has a quantization error $∆U_{T}{}^{2}$, the square, $U_{B}{}^{2}$, of the bottom electrode output voltage has a quantization error $∆U_{B}{}^{2}$, and the quantization error of the final output acceleration is ∆*a*:(18)$$a+∆a=\varepsilon A\frac{\left(U_{T}{}^{2}+∆U_{T}{}^{2}\right)d_{2}{}^{2}-\left(U_{B}{}^{2}-∆U_{B}{}^{2}\right)d_{1}{}^{2}}{md_{1}{}^{2}d_{2}{}^{2}}\;=\varepsilon A\frac{\left(B+∆B+∆U_{T}{}^{2}\right)d_{2}{}^{2}-\left(B-∆B-∆U_{B}{}^{2}\right)d_{1}{}^{2}}{md_{1}{}^{2}d_{2}{}^{2}}\;=Constant+Scale*∆B+\varepsilon A\frac{∆U_{T}{}^{2}d_{2}{}^{2}+∆U_{B}{}^{2}d_{1}{}^{2}}{md_{1}{}^{2}d_{2}{}^{2}}$$

The quantization error, ∆a, of the output acceleration can be solved as:(19)$$∆a=\frac{\varepsilon A}{m}\frac{∆U_{T}{}^{2}d_{2}{}^{2}-∆U_{B}{}^{2}d_{1}{}^{2}}{d_{1}{}^{2}d_{2}{}^{2}}=\frac{\varepsilon A}{m}\left(∆U_{T}{}^{2}\frac{1}{d_{1}{}^{2}}+∆U_{B}{}^{2}\frac{1}{d_{2}{}^{2}}\right)\;=\frac{1}{2}∆U_{T}{}^{2}\left(Scale+\frac{Constant}{B}\right)+\frac{1}{2}∆U_{B}{}^{2}\left(Scale-\frac{Constant}{B}\right)$$

According to Formula (16), the error of control quantity, ∆∆B, can be solved as:(20)$$∆∆B=\frac{∆a}{Scale}=\frac{1}{2}∆U_{T}{}^{2}\left(1+\frac{Constant}{B*Scale}\right)+\frac{1}{2}∆U_{B}{}^{2}\left(1-\frac{Constant}{B*Scale}\right)$$

The closed-loop control quantity $∆B_{out}=∆B-∆∆B=∆B-\frac{1}{2}∆U_{T}{}^{2}\left(1+CBS\right)-\frac{1}{2}∆U_{B}{}^{2}\left(1-CBS\right)$, where $CBS=\frac{Constant}{B*Scale}$. The error between the corrected output and the theoretical output is shown in Figure 8b, and the maximum error is 0.23 µg.

In principle, this method of simultaneous control of the top and bottom electrodes’ voltages realizes complete linear control when the absolute equal distance between the top and the middle electrode and that of the bottom to the middle electrode cannot be guaranteed in the practical structure. The range of the accelerometer is determined by the square of the maximum voltage available, and as a result, a relatively high range can be achieved by a low control voltage.

## 4. Implementation and Test Results

### 4.1. Test Environment Construction

Due to the test limitations, ±1 g non-linearity, bias instability, short-term bias stability, and short-term bias repeatability at room temperature have been measured in open-loop and closed-loop control.

As for the acceleration test, a test bench, as shown in Figure 9a, was built, which consists of a four-inch high-precision rotary table, and the absolute accuracy of the angle can reach 0.04°. The best accuracy of acceleration near ±1 g of gravity acceleration can be achieved as 0.3 µg, and the worst accuracy near 0 g of gravity acceleration was 700 µg. The accelerometer module was installed vertically on the horizontal base through the 3D-printed connector, and the conductive slip ring was installed on the rotary table, and then the open-loop and closed-loop tests were realized within ±1 g by using its self-weight.

In the test, the accelerometer module used the 5 V power supply provided by the USB module, and output 4-byte acceleration information and 2-byte temperature data through the designed UART or SPI. Then, UART-to-USB was used to process and store data in the computer.

### 4.2. Test Results

#### 4.2.1. Open-Loop Test

The test data were not filtered. In the open-loop test, the bias repeatability was 155 µg, which was obtained from 10 occurrences of power failure for 10 s, and the data of 10 s were acquired after powering on. As shown in Figure 9e, the Allen deviation processing was performed on the data obtained by standing for 5 h at a 1500 Hz sampling rate, and the bias instability at 5.5 s was 3.09 µg and the bias stability was 64 µg after a 1 s average. Figure 9c shows the non-linear error after linear fitting of acceleration input and voltage output under the open-loop test. The non-linearity of a single measurement was 1500 ppm.

#### 4.2.2. Closed-Loop Test

The test data were not filtered. In the closed-loop test, the bias repeatability was 12.74 µg, which was obtained from 10 occurrences of power failure for 10 s, and the data of 10 s were acquired after powering on. As shown in Figure 9f, the Allen deviation processing was performed on the data obtained by standing for 1 h at a 880 Hz sampling rate, and the bias instability at 10 s was 3.92 µg and the bias stability was 10.5 µg after a 1 s average. Figure 9d shows the non-linear error after linear fitting of acceleration input and voltage output under the closed-loop test. The non-linearity was improved to 130 ppm.

The performance test of closed-loop accelerometer is shown in Table 2, due to the angle error of the turntable itself, the non-linearity measurement of the closed-loop control was limited. However, from the existing results, the closed-loop control method proposed in this work achieved better non-linearity without modification compared to other works, as presented in Table 3. The non-linear performance was also better than the recent high-performance closed-loop capacitive accelerometer product MAXL-CL-3015-30 [25] (300 ppm).

## 5. Conclusions

In this paper, a closed-loop capacitive accelerometer using a ring-diode capacitance detection scheme and a high-linearity closed-loop control scheme was introduced, and related theoretical derivation and experimental testing were carried out. The ring-diode capacitance detection scheme applied to the parallel-plate capacitive accelerometer has the advantages of simple structure, high magnification, and without the need for a demodulation circuit and secondary amplification. The proposed high-linearity closed-loop control scheme can theoretically obtain the linear relationship between the acceleration output and the control quantity without changing the sensitive structure. It can be applied to differential capacitive accelerometers of any variable-gap capacitor to simplify the accelerometer detection and control structure and improve the linearity of the accelerometer. The test results showed that the theoretical model of the ring-diode was more accurate, and furthermore, the characters of 130 ppm non-linearity and 10.5 µg short-term bias stability for the closed-loop accelerometer have been achieved.

## Figures and Tables

**Figure 1 sensors-23-01568-f001:**
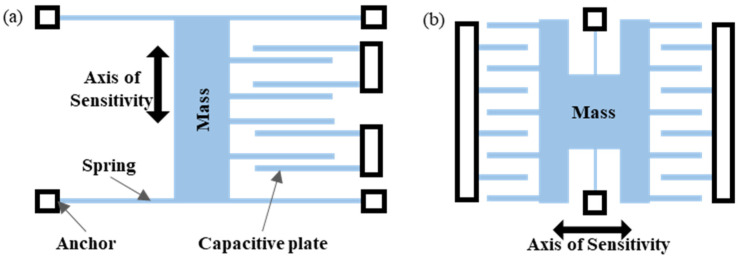
Differential capacitive accelerometer structure: (**a**) variable-gap capacitor accelerometer and (**b**) variable-area capacitive accelerometer.

**Figure 2 sensors-23-01568-f002:**
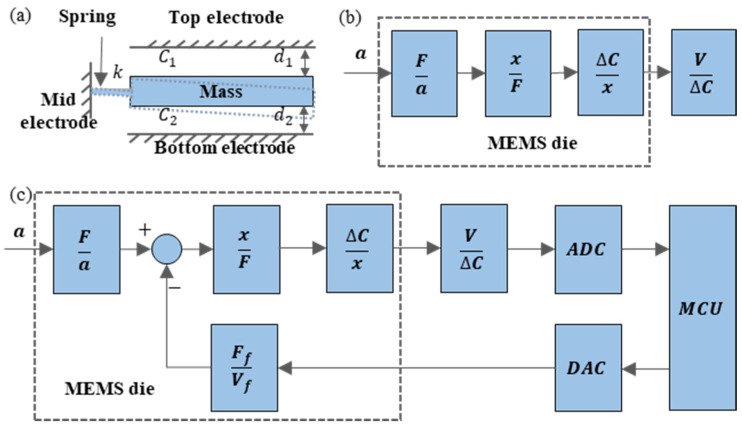
Parallel-plate capacitive accelerometer: (**a**) MEMS capacitive accelerometer structure, (**b**) acceleration-sensitive principle, and (**c**) closed-loop detection scheme of the accelerometer.

**Figure 3 sensors-23-01568-f003:**
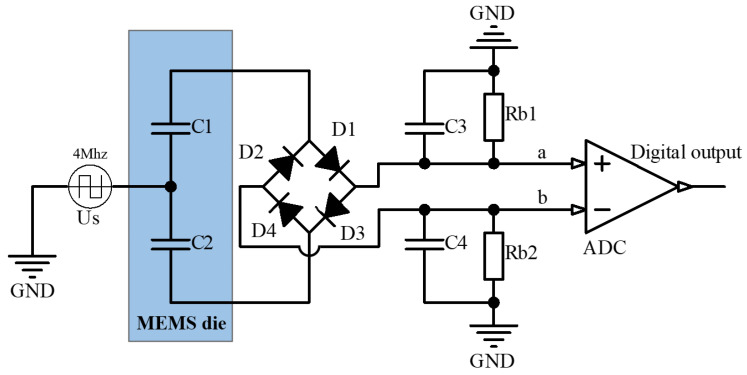
Schematic diagram of the capacitive open-loop accelerometer based on ring-diode capacitance detection.

**Figure 4 sensors-23-01568-f004:**
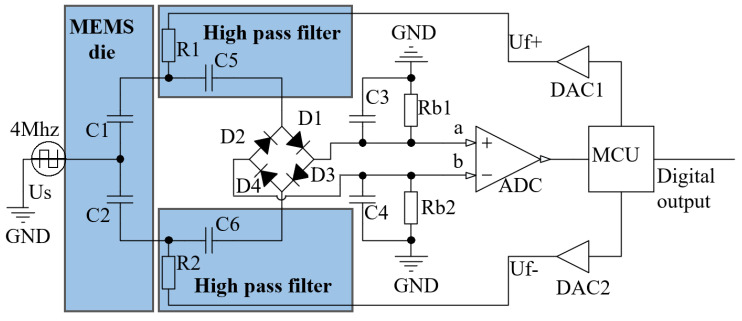
Schematic diagram of the capacitive closed-loop accelerometer based on ring-diode capacitance detection.

**Figure 5 sensors-23-01568-f005:**
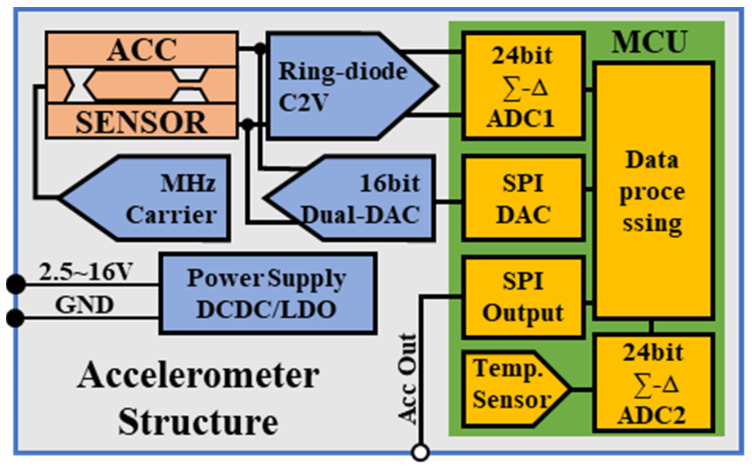
Closed-loop functional diagram.

**Figure 6 sensors-23-01568-f006:**
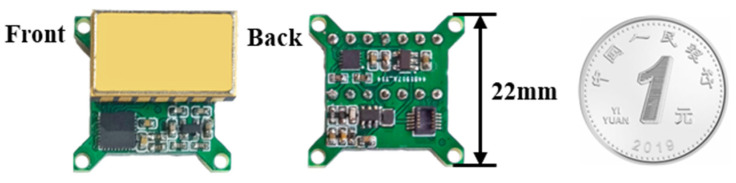
Physical drawing of the closed-loop accelerometer.

**Figure 7 sensors-23-01568-f007:**
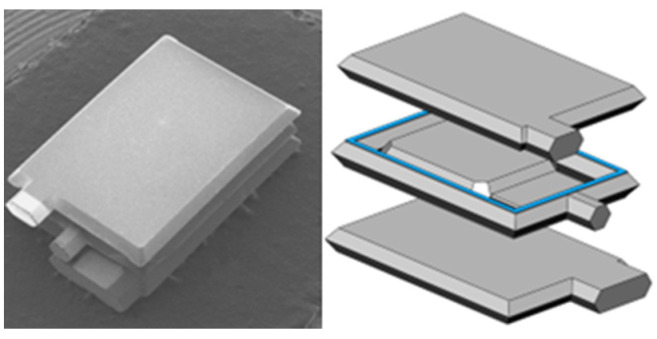
SEM and exploded views of the MEMS die.

**Figure 8 sensors-23-01568-f008:**
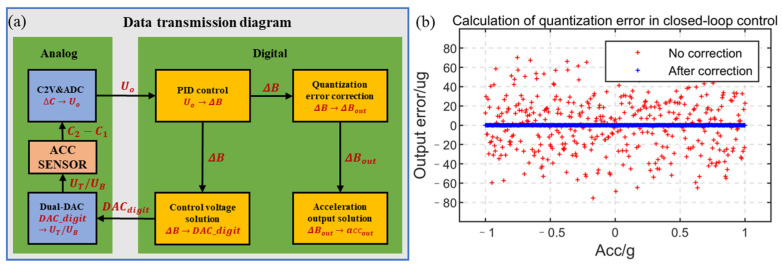
Quantization error: (**a**) data transmission diagram and (**b**) calculation of the quantization error in closed-loop control.

**Figure 9 sensors-23-01568-f009:**
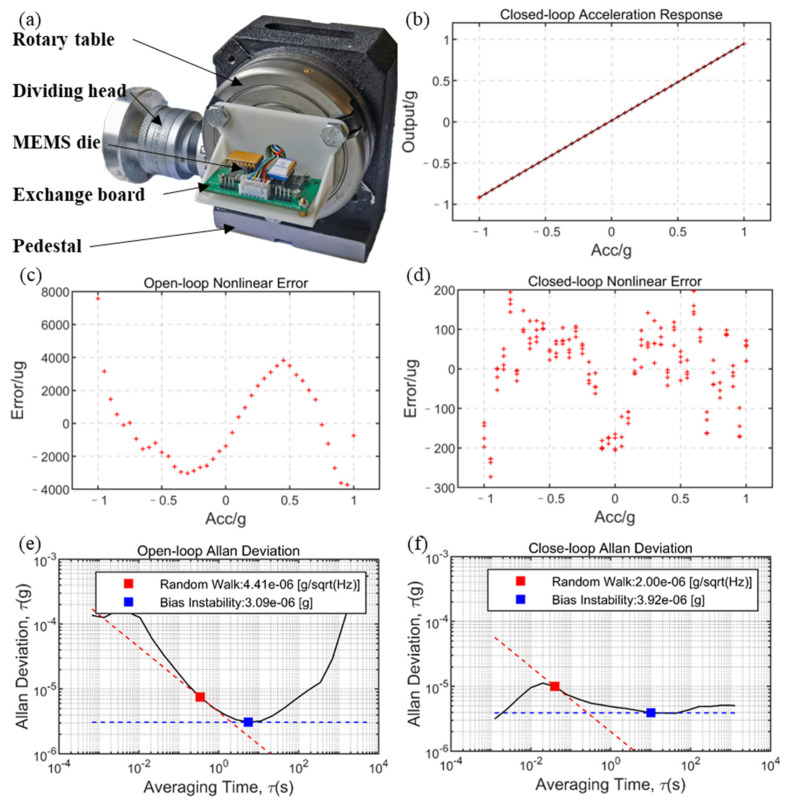
Test conditions: (**a**) test platform, (**b**) closed-loop acceleration response, (**c**) non-linear error of open-loop test, (**d**) non-linear error of closed-loop test, (**e**) Allen variance of open-loop test, and (**f**) Allen variance of closed-loop test.

**Table 1 sensors-23-01568-t001:** Closed-loop control circuit element parameters.

Element	Value/Model
C1, C2	≈32 pF
C3, C4, C5, C6	1 nF
R1, R2	100 kΩ
Rb1, Rb2	20 kΩ
D1, D2, D3, D4	HSMS-2829
MCU	MSP430I2040
DAC	DAC80502

**Table 2 sensors-23-01568-t002:** Performance statistics of the closed-loop accelerometer.

Performance	Value
Full-scale	±2 g
Non-linearity (IEEE)	130 ppm
Short-term bias stability	10.5 µg (1 s average)
Bias instability	3.92 µg (10 s)
Short-term repeatability	12.74 µg
Scale factor repeatability	11.61 ppm
Output mode	SPI/UART
Supply voltage (VDD)	2.5~16 V
Power consumption	150 mW
Size	22 × 23 mm^2^

**Table 3 sensors-23-01568-t003:** Performance comparison between recently reported closed-loop capacitive accelerometers and that in this work.

Parameter	Yunus [26] 2014	Chu [27] 2016	Dong [28] 2019	Huang [29] 2014	Li [30] 2017	Dong [28] 2019	This work
Structure	Comb-finger structure			Parallel-plate structure			
Non-linearity	1500 ppm	2220 ppm	<2000 ppm	2000 ppm	500 ppm	<5000 ppm	130 ppm

## Data Availability

Not applicable.

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
