# Peer review of "A High-Linearity Closed-Loop Capacitive Micro-Accelerometer Based on Ring-Diode Capacitance Detection"

_sensors, 2023, doi:10.3390/s23031568_

Round 1

Reviewer 1 Report

The design scheme of the circuit is relatively simple, but the performance improvement is great.The section describing the electrode control method for the "sandwich" structure is too long. It lacks of circuit simulation in "Systems Analysis" part. How to improve the linearity of the accelerometer should be described in details.

Reviewer 2 Report

The submitted article reports a close-loop accelerometer. In the opinion of the reviewer the submitted article cannot be published in this journal due to the following reasons:

- The article lacks of novelty, it is well known that close-loop mechanisms enhances and affects the performances of electrostatic-capacitive sensors in terms of sensitivity, non-linearity and measurement range. To this extent please try to explore further the literature: check for instance these articles:

DOI: 10.1016/j.sna.2020.112127 
DOI: 
10.1109/JMEMS.2012.2189361
DOI: 10.1109/CCTA.2017.8062501
DOI: 10.1016/j.sna.2017.07.048
DOI: 10.1016/j.mechatronics.2018.05.007

- The article lacks of clarity, it is really hard to understand and follow the paper. For instance, the following sentences need to be rechecked:
-"the low-frequency control signal can be applied to the same electrode which the high-frequency detection at the same time, and then the high pass processing is performed at the subsequent detection end to remove the applied low-frequency control signal, so as to realize the control and detection multiplexing of single electrode. "
-"Assuming that the circuit is stable under the action of positive half cycle or negative half cycle of square wave, the DC voltage ?? at point ?. AC amplitude under carrier action Δ??; Same as ?? and Δ??. Analyze ?3 charge and discharge, after the negative half cycle turns to positive half cycle and becomes stable, ignore the voltage rise time at point ?, the ?1 loss no charge in circuit, ?2 loss charge, ?3 supplement charge, which can be obtained from charge conservation: "
-"The acceleration is ?, the plate area is ?, the quality of proof mass is m, and the dielectric constant is ?. "
-"At the same time, only one electrode applies control voltage to the top electrode or the bottom electrode "
- "It can be seen from the above formula that under the same capacitance detection accuracy, to obtain higher acceleration detection accuracy, "

- It is not clear why the term Sandwich is used. Are the Authors referring to parallel-plate configuration? To this extent why not using variable-area electrode to solve non-linearity?

- It is not clear if the accelerometer has been fabricated or if it has been used an off-the-shelf component. No details have been provided. Figure 1a does not provide any significant information on how the MEMS device is deployed. Furthermore Figure 1a cannot be related to Eq.1 since DeltaC, d and x are not reported in the figure nor described.

- The values of the component reported in the schematic circuit of Figure 2 have not been reported in the text (R1, R2, C6, C5), the front end circuit has not been described accurately. Which diodes have been employed? How about the MCU, DAC and ADC?

- A comparison with the state-of-the-art has to be provided to validate the proposed system.

Reviewer 3 Report

There are definitely quality results in this paper, but it is not very well written. Especially, English is sometimes very hard to follow. I suggest you undertake a thorough language check, there are too many instances of bad wording for me to be able to list them here - I will emphasize just a few in this review.

Generally, I feel that you should make extensive use of graphic (figures) in your paper. Will list some of examples below.

1. Your references from the Introduction are too application specific, I suggest you mention some MEMS accelerometer groundwork references as well.

2. In formula (1) there are symbols "d" and "x", but you do not explain them neither in text nor in the Figure 1.a)

3. In line 89, I believe you should write "increased" instead of "improved"

4. I suggest you use a figure to accompany your text starting in line 92 with "At present, high-precision..." and ending in line 97 "due to DRIE."

5. Please elaborate on the meaning of DRIE in line 97

6. Again, a figure, or a reference to another, should accompany the text starting in line 107, through line 109.

7. Please add some characteristics of the actual MEMS die used, in Section 2.3

8. Your Section 3.1 need more details, I would suggest a timing diagram as well. Also, there is particularly bad English (D1 loss no charge in circuit, D2 loss charge, D3 supplement charge... this is incomprehensible)

9. In line 170, you state that the open-loop test circuit diagram is shown in Figure 2, but there is actually closed-loop circuit in the figure.

10. It would benefit an ordinary reader to explain why, in the closed loop control scheme, there is no need to take into account the accelerometer stiffness (k)

11. When estimating the quantization error in the closed loop control, and correcting for it, in Figure 5.b), how do you deduce the sign of the error and get the corrected error to behave so nicely? Please elaborate.

12. In Table 1, you mention that your Full-Scale range is +-2g, but in the tests you did +-1g. Is this a typo?

13. In line 349, I believe you wanted to type "repeatability" instead of "bias", in the basis of 12.74ug value you mention in the same line.

Round 2

Reviewer 2 Report

The article has been improved but still has flaws that should be fixed:

- I strongly suggest to use the technical term "parallel-plate configuration" instead of "sandwich configuration" and "variable-gap capacitor" instead of "variable pole pitch" in the whole paper. 
- line 123. Check the Figures number. 21 seems not correct
- Check "2.2 Closed-loop detectionand control scheme". Add a space before "and"
- In Fig.3 and 4 please consider to use the half-circle mark for unjoined wire representation to avoid wires over the diodes.
- lines 153: the sentence begins with "As shown" and refers to frequencies values but it is not possible to relate the image with the specified values. I suggest to name the signals and insert them in the figures including frequencies values. "The signal entering the ring diode" does not provide any valuable information.
- line 180: substitute "as Figure 2a shown" with "as Figure 2a shows" or "as shown in Figure 2a"
- it is still not clear if the MEMS device has been fabricated or it is a commercial component. In the first case please add an optical or SEM image of the actual structure.
- Figure 3 has to be cited in the text before Figure 4

Reviewer 3 Report

The article has been improved.

However, there are still some problems with language, which I believe will be polished during proofreading process.

Author Response

Dear Reviewers,

Thank you very much for your time involved in reviewing the manuscript and your encouraging comments on the merits.

Comments: “The article has been improved.

However, there are still some problems with language, which I believe will be polished during proofreading process.”

Thank you for your previous suggestions on our article. We have once again looked for English translation to correct the problem of English writing, hoping to improve the English writing level of this article.

We would like to take this opportunity to thank you for all your time involved and this great opportunity for us to improve the manuscript.

Sincerely,

Qi Tao